# A Newly Built Model of an Additive Stem Taper System with Total Disaggregation Model Structure for Dahurian Larch in Northeast China

Yanli Xu [1,2], Lichun Jiang [2,*] and Muhammad Khurram Shahzad [2]

1. School of Mining Engineering, Heilongjiang University of Science and Technology, Harbin 150022, China; xuyl@usth.edu.cn
2. Key Laboratory of Sustainable Forest Ecosystem Management-Ministry of Education, School of Forestry, Northeast Forestry University, Harbin 150040, China; muhammad@nefu.edu.cn
* Correspondence: jlichun@nefu.edu.cn; Tel.: +86-451-8219-0878

**Abstract:** Stem taper function is an important concept in forest growth and yield modeling, and forest management. However, the additivity of the function and the inherent correlations between stem components (diameter outside bark—dob, diameter inside bark—dib, and double-bark thickness—dbt) are seldom considered. In this paper, a total disaggregation model (TDM) structure was developed based on the well-known Kozak (2004) model to ensure the additivity of the stem components. The reconstructed model was fitted with the data of 1281 felled Dahurian larch trees from three regions of Daxing'anling Mountains in Northeast China. The results from TDM were compared with other additive model structures including adjustment in proportion (AP), non-additive taper models (NAM), and three logical structures of NSUR (AMO, SMI, SMB). The results showed that the difference was significant among the three regions. The performance of TDM was slightly better than those of other model structures. Therefore, TDM was considered as another optimal additive system to estimate stem, bark thickness, and volume predicting for Dahurian larch in Northeast China besides NSUR, a method widely used in calculating additive volume or biomass throughout the world. We believe this work is cutting-edge, and that this methodology can be applied to other tree species.

**Keywords:** additive system; stem taper; bark thickness; Dahurian larch

## 1. Introduction

The diameter outside bark (dob), inside bark (dib), and bark thickness are crucial measurements in forestry. In particular, dib is used to calculate the wood volume of logs and trees, and is usually measured by subtracting the bark thickness from the dob, whose accuracy mainly depends on the precise estimation of the bark thickness and dob. Incorrect estimation of bark thickness may lead to an inaccurate estimation of bark volume and stand volume in forest inventory, increment study or the log trade [1]. In recent years, accurate predictions of bark products have become a matter of interest for forest managers [2–5]. The utility of bark has transformed from unnecessary residue to high-value biomass energy [6]. In addition, the value of bark volume has been growing progressively, as it can be used to estimate biomass and quantify carbon stocks [7]. Therefore, new bark thickness models are being developed, and their importance has been gradually recognized by forestry scholars.

Previously, model researchers usually considered the dob, dib or bark thickness separately, and often neglected the inherent correlations among them, even if the accuracy of a single model was very high. Recently, many scholars have solved the additive problem by constructing additive models in forestry, such as additive volume, biomass or crown width equations [8–13]. In this study, we used the stem taper function to construct additive models for dob, dib and bark thickness. Through stem taper function, were able to quantify

and estimate the whole stem volume, single log volumes of any length, merchantable height in response to any diameter or from any stump height, and diameter inside/outside bark at any point of the stem [14]. Numerous taper functions have been evaluated for a wide range of tree species in different regions. In most cases, they were referenced to dbh, total height, and height from the stump to predict the diameter inside bark [15–17], and the diameter outside bark [18–21]. The bark thickness was rarely predicted in these studies [22,23].

In forestry, four methods are often used for forcing additivity of a set of the nonlinear models. These models include adjustment in proportion (AP), OLS with separating regression (OLSSR), nonlinear seemingly unrelated regression (NSUR), and total disaggregation method (TDM) [24–31]. The AP method directly partitions the total dob of a tree into two stem components—dib and double-bark thickness (dbt) by weighting, where dob, dib and dbt are estimated by OLS first. NSUR, known as a joint-generalized least square regression, proposed in the 1980s, is more flexible than AP and OLSSR [24,25], and has been widely used in additive volume or biomass equations throughout the world [32–34]. However, TDM is the best way to reflect the inherent correlations among stem components, integrating the advantages of NSUR and AP, which is also confirmed in the study.

In our study, data of 1281 harvested trees were used to establish an optimal additive system of bark thickness and stem taper (TDM) for Dahurian larch in Northeast China. The newly built system was also compared to other model structures. The additive system of stem taper was shown to provide consistent stem volume estimates for Dahurian larch with or without bark, and that the system would feasibly be a reliable method for scientific forest management.

## 2. Materials and Methods

### 2.1. Study Area and Data Collection

The study site is located to the north of Daxing'anling Mountains in the Heilongjiang and Inner Mongolia provinces of Northeast China (from 50°04′ N to 53°32′ N and from 121°50′ E to 127°00′ E). This region covers an area of 84,600 km$^2$ and is among the main areas producing high-quality wood in China. The natural secondary forest of Dahurian larch (*Larix gmelinii*) is the major forest type here. Elevation ranges from 400 to 1000 m, with a continental climate. The average annual precipitation is from 360 to 550 mm, of which 80% occurs in the summer. The average annual temperature ranges from −1.2 to −5.6 °C. The study area was assigned to three distinct regions [35]. The northwest and the southeast of the northern slope of Yilehuli Mountain are designated as regions one and two, respectively. The eastern slope of the northern part of Daxing'anling Mountain is described as region three (Figure 1).

The data used in this study were collected from 1281 felled Dahurian larch trees. Before felling, diameter at breast height (D, 1.3 m) was measured to the nearest 0.1 cm for each tree. The trees were then felled and total height (H) was measured to the nearest 0.1 m. The diameters over bark (dob) were measured at heights (h) 2, 4, 6, 8, 10, 15, 20, 30, 40, 50, 60, 70, 80, and 90% of total height, averaging 14 sections per tree (Table 1). Bark thickness was taken with the bark gauge at each relative height. The diameter inside bark (dib) was estimated with the formula: dib = dob − dbt, where dbt is double-bark thickness.

The data of the three regions were combined and randomly divided into two groups: 964 trees (75%) for model fitting and 317 trees (25%) for model validation. Summary statistics for diameter at breast height and total height of sampled trees are provided in Table 1.

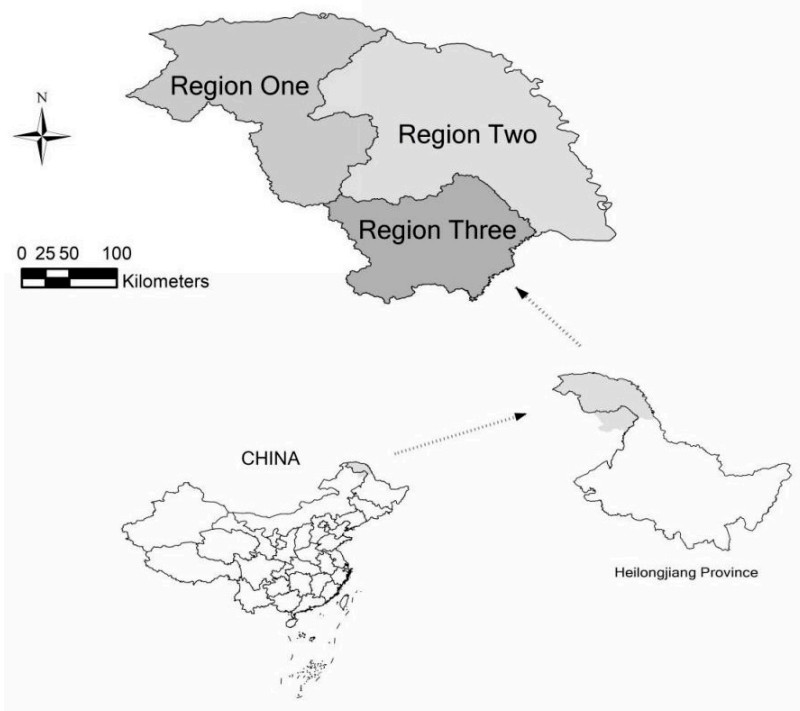

**Figure 1.** Study Area.

**Table 1.** Descriptive statistics for Dahurian larch sample trees in Daxing'anling.

|  | Group | Variables | *N* | Mean | SD | Min. | Max. |
|---|---|---|---|---|---|---|---|
| Fitting data | Region One | D (cm) | 239 | 22.55 | 10.70 | 5.1 | 62.0 |
|  |  | H (m) | 239 | 16.88 | 4.37 | 4.5 | 25.9 |
|  | Region Two | D (cm) | 282 | 26.61 | 13.08 | 5.2 | 63.4 |
|  |  | H (m) | 282 | 17.52 | 5.14 | 5.1 | 29.5 |
|  | Region Three | D (cm) | 443 | 26.17 | 13.18 | 5.4 | 61.0 |
|  |  | H (m) | 443 | 18.45 | 5.19 | 6.4 | 30.8 |
| Validation data | Region One | D (cm) | 81 | 23.16 | 12.30 | 5.5 | 50.2 |
|  |  | H (m) | 81 | 16.64 | 4.89 | 5.0 | 24.7 |
|  | Region Two | D (cm) | 94 | 26.74 | 13.83 | 5.4 | 59.0 |
|  |  | H (m) | 94 | 17.14 | 5.31 | 5.3 | 26.5 |
|  | Region Three | D (cm) | 142 | 26.36 | 13.21 | 5.5 | 54.6 |
|  |  | H(m) | 142 | 18.49 | 5.13 | 7.5 | 28.8 |

Note: *N*, number of trees; SD, standard deviation; Min., minimum; Max., maximum.

### 2.2. Base Model

To date, the stem taper function of Kozak [14] has performed well in several stem taper studies around the world [18,20,36–40]. Therefore, we selected this variable exponent function to model the stem taper in this study:

$$d = b_1 D^{b_2} H^{b_3} Q^K \tag{1}$$

$$K = b_4 T^4 + b_5 \left(\frac{1}{e}\right)^{\frac{D}{H}} + b_6 Q^m + \frac{b_7}{D} + b_8 H^{1-T^{\frac{1}{3}}} + b_9 Q$$

where *d* is stem taper; *D* is the diameter at breast height; *H* is total height; *T* = *h*/*H*, *h* is the height from the stump to any given height; *m* and $b_i$ are parameters, and *m* = 0.1;

$$Q = \frac{1 - T^{1/3}}{1 - (1.3/H)^{1/3}}$$

For Equation (1), the stem taper function of Kozak [14], various simplified forms can be generated by modifying some independent variables. Such modification can improve the model fitting precision and decrease the multicollinearity of complex models with many correlated variables. Initial model fitting results showed that Equation (1) could be simplified by removing the $H^{b_3}$ term, and the achieved model provided better fitting and lower multicollinearity for modeling dob and dib. Removal of the $b_4T^4$ and $\frac{b_7}{D}$ terms from Equation (1) further improved the fitting precision for dbt. The final modified forms are as follows:

$$Y_{dob} = f_{dob}(x,\ b) = b_1 D^{b_2} Q^{K_{dob}} + \varepsilon_{dob} \tag{2}$$

$$K_{dob} = b_3 T^4 + b_4 \left(\frac{1}{e}\right)^{\frac{D}{H}} + b_5 Q^m + b_6/D + b_7 H^{1-T^{\frac{1}{3}}} + b_8 Q$$

$$Y_{dib} = f_{dib}(x,\ b) = b_1 D^{b_2} Q^{K_{dib}} + \varepsilon_{dib} \tag{3}$$

$$K_{dib} = b_3 T^4 + b_4 \left(\frac{1}{e}\right)^{\frac{D}{H}} + b_5 Q^m + b_6/D + b_7 H^{1-T^{\frac{1}{3}}} + b_8 Q$$

$$Y_{dbt} = f_{dbt}(x,\ b) = b_1 D^{b_2} Q^{K_{dbt}} + \varepsilon_{dbt} \tag{4}$$

$$K_{dbt} = b_3 \left(\frac{1}{e}\right)^{\frac{D}{H}} + b_4 Q^m + b_5 H^{1-T^{1/3}} + b_6 Q$$

where $Y_{dob}$ and $Y_{dib}$ is the stem taper of dob and dib, $Y_{dbt}$ is the double-bark thickness of dbt, $\varepsilon_{dob}, \varepsilon_{dib}, \varepsilon_{dbt}$ is an error term of dob, dib and dbt. Other parameters are the same as aforementioned.

We introduced a dummy variable $r$ to account for stem taper difference of three regions (i.e., $r_1 = 1$ and $r_2 = 0$ for region one; $r_1 = 0$ and $r_2 = 1$ for region two; $r_1 = 0$ and $r_2 = 0$ for region three). After imposing $r$ on $b_i$, models (2)–(4) take the following forms:

$$Y_{dob} = f_{dob}(x, r,\ b) = b_1 D^{(b_2 + \lambda_1 r_2)} Q^{K_{dob}} + \varepsilon_{dob} \tag{5}$$

$$K_{dob} = (b_3 + \lambda_2 r_1 + \lambda_3 r_2) T^4 + b_4 \left(\frac{1}{e}\right)^{\frac{D}{H}} + b_5 Q^m + b_6/D + b_7 H^{1-T^{1/3}} + b_8 Q$$

$$Y_{dib} = f_{dib}(x,\ r, b) = b_1 D^{b_2} Q^{K_{dib}} + \varepsilon_{dib} \tag{6}$$

$$K_{dib} = (b_3 + \lambda_1 r_1 + \lambda_2 r_2) T^4 + b_4 \left(\frac{1}{e}\right)^{\frac{D}{H}} + (b_5 + \lambda_3 r_2) Q^m + b_6/D + b_7 H^{1-T^{1/3}} + b_8 Q$$

$$Y_{dbt} = f_{dbt}(x, r,\ b) = b_1 D^{(b_2 + \lambda_1 r_1)} Q^{K_{dbt}} + \varepsilon_{dbt} \tag{7}$$

$$K_{dbt} = b_3 \left(\frac{1}{e}\right)^{\frac{D}{H}} + (b_4 + \lambda_2 r_1) Q^m + (b_5 + \lambda_3 r_1 + \lambda_4 r_2) H^{1-T^{1/3}} + (b_6 + \lambda_5 r_1) Q$$

where $\lambda_i$ are the parameters of dummy variables. Other parameters are the same as mentioned above.

### 2.3. Total Disaggregation Method (TDM)

Tang et al. [26] proposed an additive method, disaggregation model structure. As proposed, a total model was first developed, and the estimated total model was disaggregated into components based on their proportions in the total model. The essence of this method is also NSUR. In this paper, the dob was a total model with the components of dib and dbt. This approach ensured that the sum of values of two components was equal to dob, satisfying the additive property between the total and components. The models are as follows:

$$\begin{cases} \hat{Y}_{dob} = f_{dob}(x,\ r,\ b) + \varepsilon_{dob} \\ \hat{Y}_{dib} = \frac{f_{dib}(x,r,\ b)}{f_{dib}(x,r,\ b) + f_{dbt}(x,\ r,b)} \times f_{dob}(x,r,\ b) + \varepsilon_{dib} \\ \hat{Y}_{dbt} = \frac{f_{dbt}(x,\ r,b)}{f_{dib}(x,r,\ b) + f_{dbt}(x,\ r,b)} \times f_{dob}(x,r,\ b) + \varepsilon_{dbt} \end{cases} \tag{8}$$

Based on the above model, the dob was separately fitted by nonlinear OLS. The dib and dbt were estimated by disaggregating the total model into components based on their proportions, through deducing and combining like terms, respectively, and using the NSUR method. The formulas took the following forms:

$$\begin{cases} \hat{Y}_{dob} = b_1 D^{(b_2 + \lambda_1 r_2)} Q^{K_{dob}} + \varepsilon_{dob} \\ \hat{Y}_{dib} = \hat{Y}_{dob} \big/ \left( 1 + b_1 D^{(b_2 + \lambda_1 r_1)} Q^{K_{dib}} \right) + \varepsilon_{dib} \\ \hat{Y}_{dbt} = \hat{Y}_{dob} \big/ \left( 1 + \frac{1}{b_1} D^{(-b_2 - \lambda_1 r_1)} Q^{K_{dbt}} \right) + \varepsilon_{dbt} \end{cases} \tag{9}$$

$$K_{dob} = (b_3 + \lambda_2 r_1 + \lambda_3 r_2) T^4 + b_4 \left( \frac{1}{e} \right)^{\frac{D}{H}} + b_5 Q^m + b_6 / D + b_7 H^{1-T^{1/3}} + b_8 Q$$

$$K_{dib} = (b_3 + \lambda_2 r_1 + \lambda_3 r_2) T^4 + b_4 \left( \frac{1}{e} \right)^{\frac{D}{H}} + \left( b_5 + \lambda_4 r_1 + \lambda_5 r_2 \right) Q^m + b_6 / D + \left( b_7 + \lambda_6 r_1 + \lambda_7 r_2 \right) H^{1-T^{1/3}} + \left( b_8 + \lambda_8 r_1 \right) Q$$

$$K_{dbt} = -(b_3 + \lambda_2 r_1 + \lambda_3 r_2) T^4 - b_4 \left( \frac{1}{e} \right)^{\frac{D}{H}} - \left( b_5 + \lambda_4 r_1 + \lambda_5 r_2 \right) Q^m - b_6 / D - \left( b_7 + \lambda_6 r_1 + \lambda_7 r_2 \right) H^{1-T^{1/3}} - \left( b_8 + \lambda_8 r_1 \right) Q$$

where parameters are the same as described before.

### 2.4. Adjustment in Proportion (AP)

Adjustment in proportion method is another additive method proposed by Tang et al. [26], which ensures that the sum of values of trees components is equal to dob. For example, when an indicator variable r was considered, base models (5)–(7) were separately fitted by nonlinear OLS for dob, dib and dbt. There are many similarities between AP and TDM. We made a comparison, firstly, and chose the best method to compare with other model structures. The estimates of dib and dbt were calculated as follows:

$$\hat{Y}_{dob} = f_{dob}\left( x,\ r, \hat{b}_{dob} \right) + \varepsilon_{dob}$$
$$\hat{Y}_{dib} = \frac{f_{dib}\left( x,r,\hat{b}_{dib} \right)}{f_{dib}\left( x,r,\hat{b}_{dib} \right) + f_{dbt}\left( x,\ r,\hat{b}_{dbt} \right)} \times f_{dob}\left( x,r,\ \hat{b}_{dob} \right) + \varepsilon_{dib} \tag{10}$$
$$\hat{Y}_{dbt} = \frac{f_{dbt}\left( x,\ r,\hat{b}_{dbt} \right)}{f_{dib}\left( x,r,\hat{b}_{dib} \right) + f_{dbt}\left( x,r,\hat{b}_{dbt} \right)} \times f_{dob}\left( x,r,\ \hat{b}_{dob} \right) + \varepsilon_{dbt}$$

where $\hat{Y}_{dob}$, $\hat{Y}_{dib}$, $\hat{Y}_{dbt}$ are estimates of $Y_{dob}$, $Y_{dib}$, $Y_{dbt}$, respectively. $\hat{b}_{dob}$, $\hat{b}_{dib}$, $\hat{b}_{dbt}$ are estimates of the parameter vectors obtained by fitting models (5)–(7) for dob, dib and dbt, respectively.

### 2.5. Other Model Structures to Compare

2.5.1. Non-Additive Taper Models (NAM)

The dib taper and dbt were separately fitted by OLS, and dob taper estimation was obtained by adding the two components.

$$Y_{dob} = f_{dib}(x,r,\ b) + f_{dbt}(x,\ r,b) + \varepsilon_{dob} \tag{11}$$

2.5.2. Additive Taper Models with Dob Constraint (AMO)

Following the model structure, total amount control method as specified by Parresol [25], the dib taper function and dbt with NSUR approach were constrained to equal the dob taper equation as follows:

$$\begin{cases} Y_{dib} = f_{dib}(x,r,\ b) + \varepsilon_{dib} \\ Y_{dbt} = f_{dbt}(x,r,\ b) + \varepsilon_{dbt} \\ Y_{dob} = f_{dib}(x,r,\ b) + f_{dbt}(x,r,\ b) + \varepsilon_{dob} \end{cases} \tag{12}$$

### 2.5.3. Subtraction Taper Models with Dib Constraint (SMI)

First logistic transformation of the total amount control method, the dib taper equation had the difference of simultaneous estimations of dob taper function and dbt taper equation with those of the NSUR approach. The formulas are as follows:

$$\begin{cases} Y_{dob} = f_{dob}(x,r,\ b) + \varepsilon_{dob} \\ Y_{dbt} = f_{dbt}(x,\ r,b) + \varepsilon_{dbt} \\ Y_{dib} = f_{dob}(x,r,\ b) - f_{dbt}(x,r,b) + \varepsilon_{dib} \end{cases} \tag{13}$$

### 2.5.4. Subtraction Taper Models with Dbt Constraint (SMB)

Second logistic transformation of total amount control method, the dbt had the difference of simultaneous estimations of dob taper function and dib taper equation with those of the NSUR approach. The formulas are as follows:

$$\begin{cases} Y_{dob} = f_{dob}(x,\ r,b) + \varepsilon_{dob} \\ Y_{dib} = f_{dib}(x,\ r,b) + \varepsilon_{dib} \\ Y_{dbt} = f_{dob}(x,r,\ b) - f_{dib}(x,r,\ b) + \varepsilon_{dbt} \end{cases} \tag{14}$$

where parameters are the same as aforementioned.

The model residuals of stem taper data often show heteroscedasticity. To overcome the problem, we chose $\sqrt{1/D^k}$ as the weight factor (where: $k$ was determined for each model). This factor was multiplied and programmed in the PROC MODEL procedure [41] by specifying $resid.D_i = resid.D_i / \sqrt{D_i^k}$ (where: $resid.D_i$ is the model residual of $D_i$). The heteroscedasticity was obvious in the models of dbt which was corrected accordingly. For the models of dob and dib, it was almost absent.

### 2.6. Model Assessment and Evaluation

Initially, the accuracy of the single estimate models with and without dummy variables was assessed with the fitting dataset. Later, the predictive abilities of TDM models and other models were evaluated with both fitting and validation dataset. Five statistical indexes, i.e., adjusted coefficient of determination ($R_a^2$), mean error ($\bar{e}$), residual variance ($\delta$), root mean square error ($RMSE$) and total relative error ($TRE$), were tested. The notations for these indexes are as follows:

$$R_a{}^2 = 1 - \left(1 - R^2\right)\left(\frac{N-1}{N-p}\right) \tag{15}$$

$$\bar{e} = \sum_{i=1}^{n}(Y_i - \hat{Y}_i)/N \tag{16}$$

$$\delta = \sum_{i=1}^{n}\left(Y_i - \hat{Y}_i\right)^2 / (N-1) \tag{17}$$

$$RMSE = \sqrt{\bar{e}^2 + \delta} \tag{18}$$

$$TRE = 100 \times \frac{\sum_{i=1}^{n}(Y_i - \hat{Y}_i)}{\sum_{i=1}^{n}\hat{Y}_i} \tag{19}$$

where $Y_i$ is the observation data, $\hat{Y}_i$ is the predicted value, $N$ is the total number of observations, and $p$ is the number of parameters.

The condition number ($CN$) was used to test the multicollinearity. The criteria for a $CN$ value that indicated the extent of multicollinearity were arbitrary. It was considered

that no collinearity existed in the model when the *CN* value was less than 30. The value of *CN* between 30 and 100 signified the existence of multicollinearity, however the model was still acceptable [42].

### 2.7. Ranking of Models

The ranking method proposed by Poudel and Cao [43] was used to obtain the relative and exact position of different models. The relative rank of model *i* is defined as:

$$Rank_i = 1 + \frac{(m-1)(S_i - S_{min})}{S_{max} - S_{min}} \tag{20}$$

where: $Rank_i$ is the relative rank of model *i* (*i* = 1, 2, ... , m), $S_i$ is the goodness-of-fit statistics, including $R_a{}^2$, $\bar{e}$, $\delta$, *RMSE* and *TRE*, produced by model *i*. $S_{min}$ is the minimum value of $S_i$, and $S_{max}$ is the maximum value of $S_i$. Relative ranks of 1 and m indicate the best and the worst model. Since the magnitude and the order of the $S_i$'s both are included in the ranking system, it should provide additional information than the conventional ordinal ranks.

## 3. Results

### 3.1. Fitting the Base Model

Fitting statistics for dob, dib and dbt taper equations with and without dummy variables are displayed in Table 2. The *CN* values of models (2)–(4) were less than 40, indicating less or no multicollinearity in models. As expected, the models with dummy variables (5)–(7) reflected some degree of variation and outperformed models (2)–(4). The values of *RMSE*, $R_a{}^2$, $\bar{e}$ and $\delta$ of the models with dummy variables were superior to those of the models without dummy variables. Based on three optimal models, additive systems were constructed in the following section.

**Table 2.** Goodness-of-fit statistics of single estimate models with or without dummy variables.

| Variable | Models | *RMSE* | $R_a{}^2$ | $\bar{e}$ | $\delta$ | *AIC* | *CN* |
|---|---|---|---|---|---|---|---|
| dob | Model (2) | 1.7896 | 0.9770 | −0.0041 | 3.2027 | 7333 | 35 |
| | Model (5) | 1.7624 | 0.9777 | −0.0033 | 3.1062 | 7141 | |
| dib | Model (3) | 1.9199 | 0.9675 | −0.0020 | 3.6859 | 8216 | 36 |
| | Model (6) | 1.8800 | 0.9689 | −0.0008 | 3.5344 | 7952 | |
| dbt | Model (4) | 1.0025 | 0.6456 | 0.2160 | 0.9584 | 54 | 21 |
| | Model (7) | 0.9910 | 0.6535 | 0.2125 | 0.9370 | −91 | |

Note: Models (2)–(4) are reduced models; Models (5)–(7) contain dummy variables.

### 3.2. Model Fitting for Six Methods

The six additive methods (Equations (9)–(14)) were constructed based on models (5)–(7). The goodness-of-fit statistics of six methods are shown in Table 3. Both total disaggregation methods (TDM, AP) provided the same results for dob and both of them were estimated by OLS. The same values of $R_a{}^2$ were rendered for dib. However, there was a marginal difference in the values of *RMSE*, $\bar{e}$, and $\delta$ between the methods. As per average rankings of the methods, the TDM appeared to be more attractive than AP for dib and dbt models.

Among the six models, the dob of TDM/AP, dib of NAM, and dbt of SMB were superior to the others. However, the dbt of SMI, and dob and dib of SMB, behaved inadequately while fitting. This was reflected by the average of total ranks. In general, TDM still performed slightly better than those with other methods (i.e., dib of TDM was slightly better than that of AP and SMB, dbt of TDM was second only to that of SMB). In this case, the TDM was slightly superior to the AMO, which has been widely used, in *RMSE*, $\bar{e}$, $\delta$, *TRE* and $R_a{}^2$ by 1.8, 6.5, 3.1, 3.9 and 1.6%, respectively.

**Table 3.** Goodness of fit statistics of five additive models.

| Methods | Variable | *RMSE* | $R_a{}^2$ | $\bar{e}$ | $\delta$ | $\overline{rank}$ |
|---|---|---|---|---|---|---|
| TDM | dob | 1.7624 | 0.9777 | −0.0033 | 3.1062 | 1.02 |
| | dib | 1.9004 | 0.9685 | −0.1981 | 3.5722 | 4.38 |
| | dbt | 0.9730 | 0.6652 | 0.2034 | 0.9053 | 3.04 |
| AP | dob | 1.7624 | 0.9777 | −0.0033 | 3.1062 | 1.02 |
| | dib | 1.9022 | 0.9685 | −0.1961 | 3.5801 | 4.57 |
| | dbt | 0.9818 | 0.6574 | 0.1928 | 0.9268 | 3.79 |
| NAM | dob | 1.7926 | 0.9773 | 0.2117 | 3.1685 | 4.55 |
| | dib | 1.8800 | 0.9689 | −0.0008 | 3.5344 | 1.00 |
| | dbt | 0.9910 | 0.6535 | 0.2125 | 0.9370 | 5.28 |
| AMO | dob | 1.7627 | 0.9777 | 0.0533 | 3.1042 | 1.34 |
| | dib | 1.8986 | 0.9685 | −0.1642 | 3.5776 | 4.17 |
| | dbt | 0.9909 | 0.6545 | 0.2175 | 0.9345 | 5.37 |
| SMI | dob | 1.7668 | 0.9776 | 0.0668 | 3.1173 | 1.84 |
| | dib | 1.8895 | 0.9688 | −0.1585 | 3.5449 | 2.75 |
| | dbt | 0.9950 | 0.6527 | 0.2253 | 0.9392 | 6.00 |
| SMB | dob | 1.7970 | 0.9768 | 0.0542 | 3.2263 | 5.08 |
| | dib | 1.9074 | 0.9681 | −0.1476 | 3.6165 | 5.68 |
| | dbt | 0.9582 | 0.6756 | 0.2018 | 0.8774 | 1.35 |

Note: $\overline{rank}$ is the average rank of *RMSE*, $R_a{}^2$, $\bar{e}$, and $\delta$.

Figure 2 displays a set of the residual graphs to visually present the fitting effects of additive systems for dob, dib and dbt with TDM methods. The residuals were randomly distributed and the data points did not show error trends. The parameter estimation for the taper equation systems of the six methods are listed in Table 4. All parameters were significant ($p < 0.0001$), including the estimates of dummy variables $\lambda_i$, which indicated that there was a significant difference in the taper equation systems among the three regions.

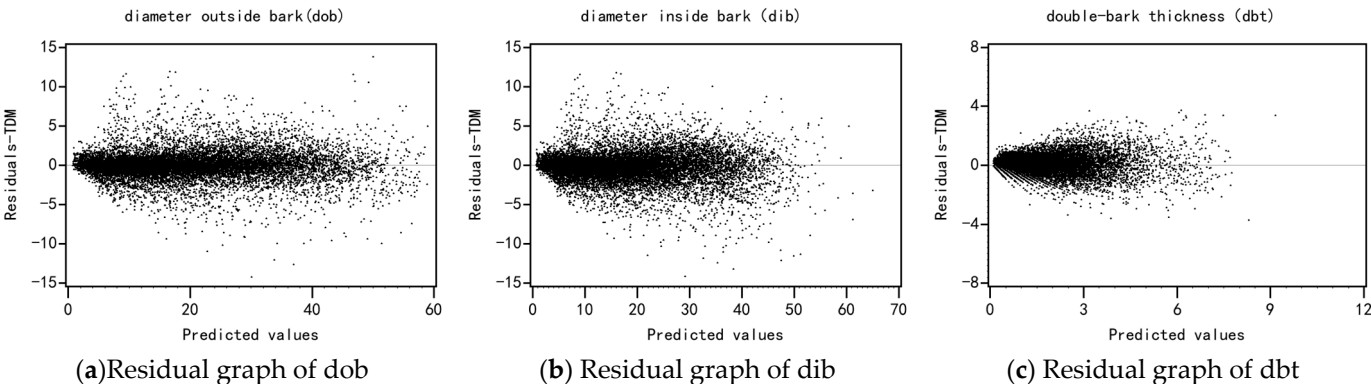

(**a**) Residual graph of dob     (**b**) Residual graph of dib     (**c**) Residual graph of dbt

**Figure 2.** Residual graphs of an additive system for dob, dib and dbt with TDM methods.

**Table 4.** Parameter estimate of the additive system with all methods.

| Para | TDM | | | AP | | | NAM | | AMO | | SMI | | SMB | |
|---|---|---|---|---|---|---|---|---|---|---|---|---|---|---|
| | dob | dib | dbt | dob | dib | dbt | dib | dbt | dib | dbt | dob | dbt | dob | dib |
| $b_1$ | 1.0766 | 0.1639 | 6.1013 | 1.0766 | 0.9027 | 0.1581 | 0.9027 | 0.1581 | 0.9642 | 0.0882 | 1.0539 | 0.0873 | 1.0474 | 0.952 |
| | (0.0072) | (0.0051) | (0.0051) | (0.0072) | (0.0073) | (0.0043) | (0.0073) | (0.0043) | (0.0073) | (0.002) | (0.0071) | (0.0019) | (0.007) | (0.0072) |
| $b_2$ | 0.9744 | −0.1301 | 0.1301 | 0.9744 | 0.992 | 0.8454 | 0.992 | 0.8454 | 0.9746 | 1.0265 | 0.9792 | 1.0267 | 0.9848 | 0.978 |
| | (0.0019) | (0.0093) | (0.0093) | (0.0019) | (0.0023) | (0.0081) | (0.0023) | (0.0081) | (0.0022) | (0.0065) | (0.0019) | (0.0065) | (0.0019) | (0.0022) |
| $b_3$ | 0.4364 | 0.0793 | −0.0793 | 0.4364 | 0.4662 | −0.7029 | 0.4662 | −0.7029 | 0.4516 | −1.2784 | 0.4398 | −1.2595 | 0.4627 | 0.5169 |
| | (0.0104) | (0.0624) | (0.0624) | (0.0104) | (0.0125) | (0.0668) | (0.0125) | (0.0668) | (0.0113) | (0.0481) | (0.0104) | (0.0475) | (0.0104) | (0.0114) |
| $b_4$ | −0.4433 | −0.2892 | 0.2892 | −0.4433 | −0.4706 | 0.7683 | −0.4706 | 0.7683 | −0.4157 | 0.9853 | −0.466 | 0.9658 | −0.4286 | −0.4073 |
| | (0.0252) | (0.1284) | (0.1284) | (0.0252) | (0.0293) | (0.0285) | (0.0293) | (0.0285) | (0.0268) | (0.0224) | (0.0251) | (0.0223) | (0.0249) | (0.0269) |
| $b_5$ | 0.3163 | 0.0221 | −0.0221 | 0.3163 | 0.3102 | 0.2534 | 0.3102 | 0.2534 | 0.3069 | 0.2943 | 0.3161 | 0.3026 | 0.3106 | 0.2793 |
| | (0.0054) | (0.034) | (0.034) | (0.0054) | (0.0066) | (0.0134) | (0.0066) | (0.0134) | (0.0061) | (0.0103) | (0.0054) | (0.0104) | (0.0054) | (0.006) |
| $b_6$ | 2.0128 | 0.2662 | −0.2662 | 2.0128 | 2.1943 | −0.6483 | 2.1943 | −0.6483 | 2.1083 | −0.5641 | 2.0604 | −0.6503 | 1.8789 | 2.0452 |
| | (0.1427) | (0.0170) | (0.0170) | (0.1427) | (0.168) | (0.0632) | (0.168) | (0.0632) | (0.1527) | (0.0507) | (0.1423) | (0.051) | (0.1402) | (0.1535) |
| $b_7$ | 0.1013 | −0.2024 | 0.2024 | 0.1013 | 0.0809 | − | 0.0809 | − | 0.0735 | − | 0.1043 | − | 0.1019 | 0.0711 |
| | (0.0033) | (0.0841) | (0.0841) | (0.0033) | (0.004) | | (0.004) | | (0.0038) | | (0.0033) | | (0.0033) | (0.0037) |
| $b_8$ | −0.2882 | 1.9610 | −1.9610 | −0.2882 | −0.2713 | − | −0.2713 | − | −0.2671 | − | −0.2926 | − | −0.2700 | −0.2300 |
| | (0.0217) | (0.5210) | (0.5210) | (0.0217) | (0.0259) | | (0.0259) | | (0.0241) | | (0.0216) | | (0.0215) | (0.0241) |
| $\lambda_1$ | −0.1488 | 0.0478 | −0.0478 | −0.1488 | −0.1769 | 0.0401 | −0.1769 | 0.0401 | −0.1664 | 0.0544 | 0.003 | 0.0548 | −0.1529 | −0.1754 |
| | (0.0092) | (0.0034) | (0.0034) | (0.0092) | (0.0105) | (0.0031) | (0.0105) | (0.0031) | (0.0096) | (0.0024) | (0.0004) | (0.0025) | (0.0092) | (0.0099) |
| $\lambda_2$ | −0.1433 | 0.6391 | −0.6391 | −0.1433 | −0.1076 | 0.2576 | −0.1076 | 0.2576 | −0.1049 | 0.2809 | −0.1517 | 0.2675 | −0.1873 | −0.2197 |
| | (0.0082) | (0.083) | (0.083) | (0.0082) | (0.0122) | (0.0388) | (0.0122) | (0.0388) | (0.0111) | (0.0278) | (0.0091) | (0.0283) | (0.008) | (0.0088) |
| $\lambda_3$ | 0.0012 | 0.6082 | −0.6082 | 0.0012 | −0.0402 | −0.0758 | −0.0402 | −0.0758 | −0.0302 | −0.0805 | −0.1419 | −0.0815 | −0.0082 | 0.0166 |
| | (0.0004) | (0.0636) | (0.0636) | (0.0004) | (0.004) | (0.0254) | (0.004) | (0.0254) | (0.0037) | (0.0183) | (0.0082) | (0.0187) | (0.0002) | (0.0012) |
| $\lambda_4$ | − | 0.8942 | −0.8942 | − | − | 0.0469 | − | 0.0469 | − | 0.0226 | − | 0.0355 | − | − |
| | | (0.0623) | (0.0623) | | | (0.0062) | | (0.0062) | | (0.0047) | | (0.0048) | | |
| $\lambda_5$ | − | −0.9709 | 0.9709 | | | | − | 0.3403 | − | 0.2884 | − | 0.4213 | − | − |
| | | (0.0488) | (0.0488) | | | | | (0.1128) | | (0.0864) | | (0.0867) | | |
| $\lambda_6$ | − | −0.1525 | 0.1525 | | | | − | − | − | − | − | − | − | − |
| | | (0.0305) | (0.0305) | | | | | | | | | | | |
| $\lambda_7$ | − | −0.2156 | 0.2156 | | | | − | − | − | − | − | − | − | − |
| | | (0.0148) | (0.0148) | | | | | | | | | | | |
| $\lambda_8$ | − | −0.1131 | 0.1131 | | | | − | − | − | − | − | − | − | − |
| | | (0.1627) | (0.1627) | | | | | | | | | | | |

Note: Approximate SEs appear in parentheses. dob: diameter outside bark. dib: diameter inside bark. dbt: double-bark thickness. $b_i$: parameters to estimate. $\lambda_i$: Parameters of dummy variables.

*3.3. Model Validation for Six Methods*

Table 5 shows the slight superiority of TDM, compared with other model structures. The methods of NAM, AMO, SMI and SMB differed slightly, but were less accurate. The prediction precision of TDM was higher than these four methods as evident from the average ranks. The precision of TDM was closely followed by AP for all variables. The dob and dib were overestimated by TDM model structure. In contrast, they were underestimated by the NAM model. The dob was overestimated and the dib was underestimated consistently by AMO, SMI and SMB models. All model structures underestimated dbt. The differences in mean error ($\bar{e}$) of dob, dib and dbt of each model structure reveals the essence of additivity.

**Table 5.** Evaluation indices produced from TDM and other compared model structures.

| Methods | Variable | $\bar{e}$ | $\delta$ | *RMSE* | *TRE* | $\overline{rank}$ |
|---|---|---|---|---|---|---|
| TDM | dob | −0.0010 | 3.6637 | 1.9141 | 18.6417 | 1.08 |
|  | dib | −0.1684 | 3.9716 | 2.000 | 22.1320 | 2.25 |
|  | dbt | 0.1834 | 0.8734 | 0.9524 | 51.6132 | 1.60 |
| AP | dob | −0.0010 | 3.6637 | 1.9141 | 18.6417 | 1.08 |
|  | dib | −0.1653 | 3.9909 | 2.0046 | 22.2437 | 2.54 |
|  | dbt | 0.1718 | 0.8961 | 0.9621 | 52.5948 | 1.69 |
| NAM | dob | 0.2134 | 3.7208 | 1.9407 | 19.1411 | 3.68 |
|  | dib | 0.0240 | 4.0625 | 2.0157 | 22.8840 | 2.54 |
|  | dbt | 0.1893 | 0.8975 | 0.9661 | 53.2283 | 2.83 |
| AMO | dob | 0.0510 | 3.6751 | 1.9177 | 18.7497 | 1.63 |
|  | dib | −0.1371 | 4.1353 | 2.0382 | 23.0849 | 4.67 |
|  | dbt | 0.1881 | 0.9710 | 1.0032 | 57.5426 | 5.35 |
| SMI | dob | 0.0684 | 3.6535 | 1.9126 | 18.6558 | 1.42 |
|  | dib | −0.1276 | 4.1218 | 2.0342 | 23.0217 | 4.39 |
|  | dbt | 0.1960 | 0.9754 | 1.0069 | 58.0759 | 6.00 |
| SMB | dob | 0.0553 | 3.8947 | 1.9743 | 19.8742 | 5.08 |
|  | dib | −0.1225 | 4.2055 | 2.0544 | 23.4959 | 5.70 |
|  | dbt | 0.1777 | 0.8750 | 0.9522 | 51.5387 | 1.31 |

Note: $\overline{rank}$ is the average ranks of $\bar{e}$, $\delta$, *RMSE* and *TRE*.

To evaluate the predictive ability of each method across the entire stem, the relative height was divided into nine sections. The six methods were further assessed on the basis of graphical analysis (Figures 3–5). The three figures show the $\bar{e}$, $\delta$, *RMSE*, and *TRE* for the six model structures across relative height classes for dob, dib and dbt.

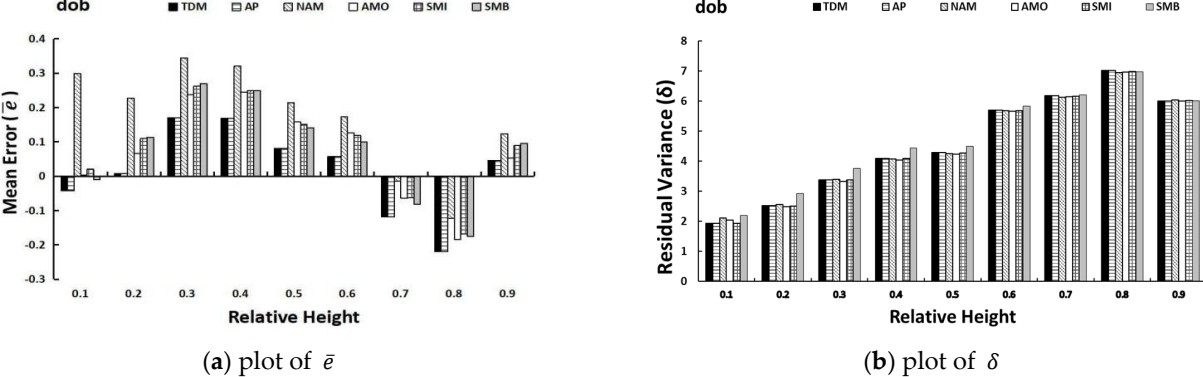

(**a**) plot of $\bar{e}$      (**b**) plot of $\delta$

**Figure 3.** *Cont.*

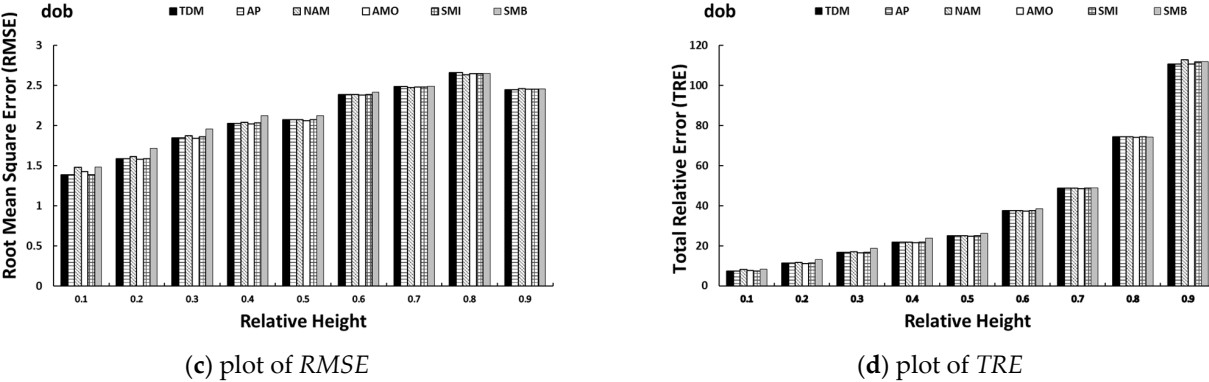

(**c**) plot of *RMSE*　　　　　　　　　　　　(**d**) plot of *TRE*

**Figure 3.** Plots of evaluation statistics from six methods against relative height classes for dob.

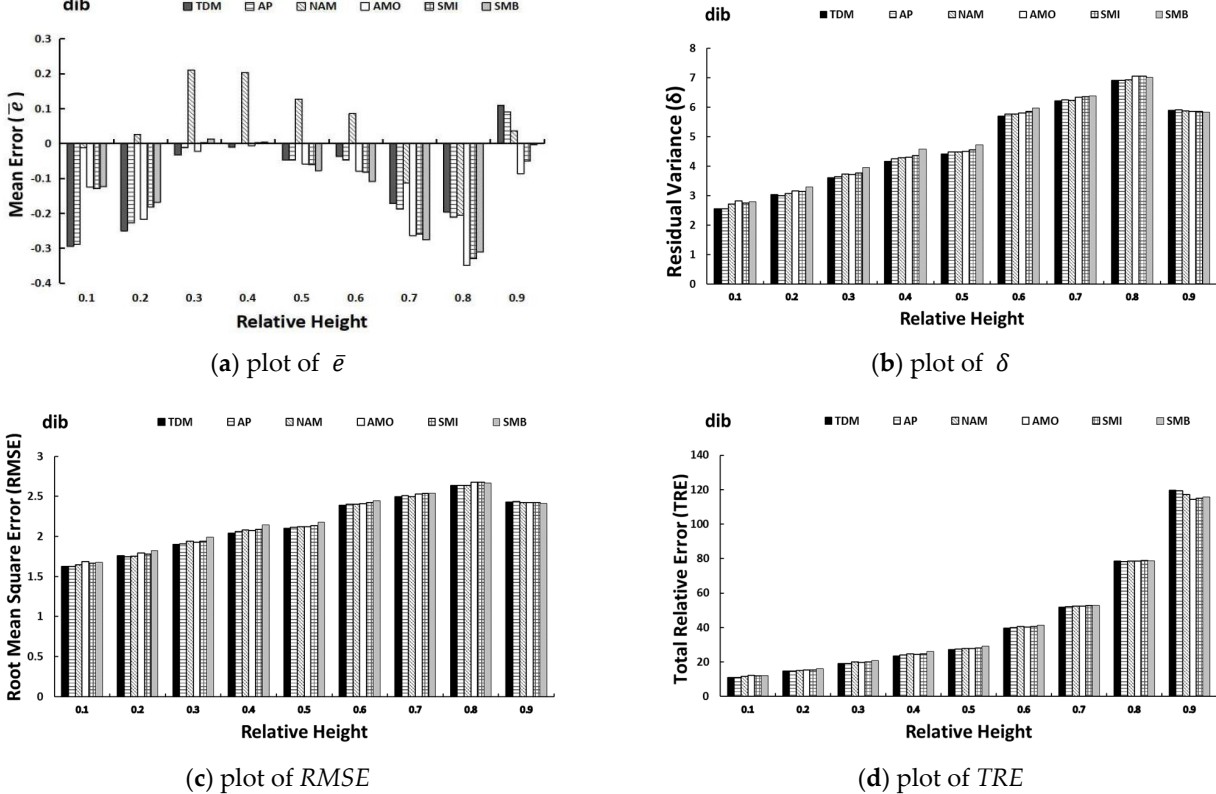

(**a**) plot of $\bar{e}$　　　　　　　　　　　　(**b**) plot of $\delta$

(**c**) plot of *RMSE*　　　　　　　　　　　　(**d**) plot of *TRE*

**Figure 4.** Plots of evaluation statistics from six methods against relative height classes for dib.

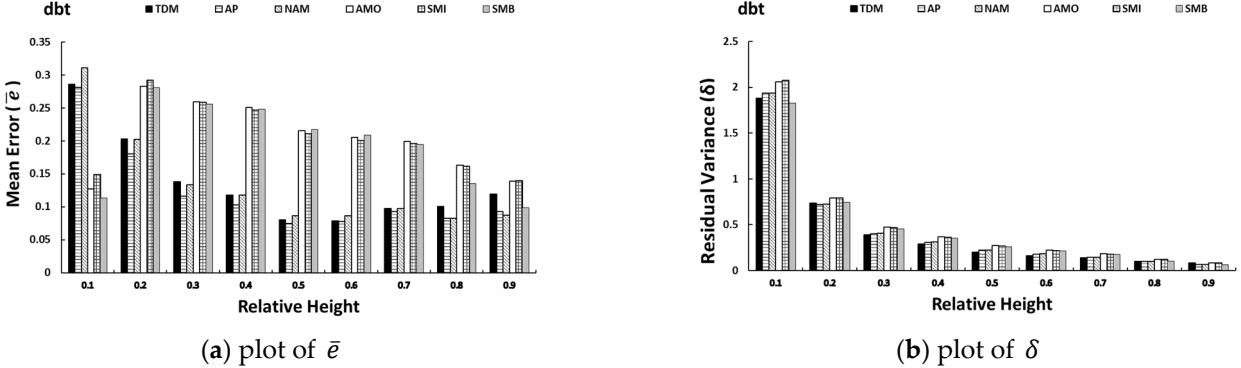

(**a**) plot of $\bar{e}$　　　　　　　　　　　　(**b**) plot of $\delta$

**Figure 5.** *Cont.*

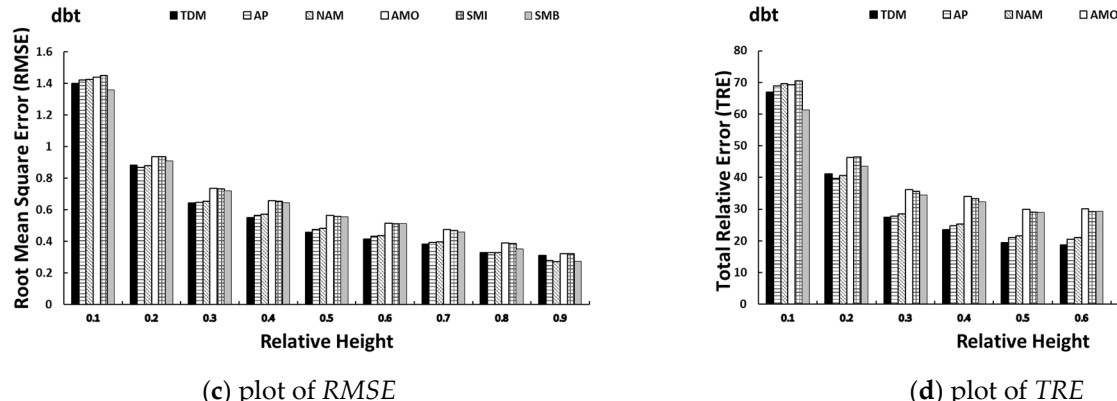

(**c**) plot of *RMSE*    (**d**) plot of *TRE*

**Figure 5.** Plots of evaluation statistics from six methods against relative height classes for dbt.

Figure 3 shows that additive models of dob with TDM and AP methods slightly outperformed other models in terms of *δ*, *RMSE*, and *TRE* for most relative height classes, with the exception of relative height classes (rh = 0.7 and rh = 0.8), which could be attributed to the base of live crown. The change of tapering grades at the base of the live crown may result in the defective performance of taper models. The results were in accordance with those of Lee et al. [44] and Rodríguez et al. [23]. All methods underestimated the dob (0.2 = < rh < 0.6 and rh = 0.9) and exhibited the smallest residuals.

Figure 4 shows that the additive models of dib with TDM and AP slightly outperformed other methods in terms of *δ*, *RMSE*, and *TRE* for most relative height classes, with the exception of relative height classes (rh = 0.2 and rh = 0.9). Most additive models with six methods nearly overestimated the dib, with the exception of relative height classes (0.2 = < rh < 0.6) for the model with the NAM method, relative height classes (0.3 = < rh <0.4) for the models with SMI and SMB methods, and relative height classes (rh = 0.9) for the models with TDM, AP and NAM methods.

Figure 5 shows that additive models of dbt with the TDM method slightly outperformed other models in terms of *δ*, *RMSE*, and *TRE* for most relative height classes (0.3 = < rh < 0.8), and the model with the SMB method also outperformed other models in terms of *ē*, *δ*, *RMSE*, and *TRE* for lower section (rh = 0.1). All model structures underestimated values of dbt. These conclusions are consistent with Table 5.

The six methods were subsequently used to describe the stem tapers in the three regions (Figure 6), each graph describing trees of the same size (i.e., same D and H), representing the average tree. There were no great differences, with graphs showing that all stem tapers for dob and dib were almost identical, and that these function plots had consistent trends, except for a slight difference in the lower section among the three regions. However, the graphs showed differences in the dbt curves across the six methods, and the predicted values of SMB for dbt were relatively smaller than those of other methods in region two at the interval (0.4 ≤ rh). The predicted values of TDM for dbt were relatively bigger to those of other methods in region three at the interval (0.5 ≤ rh ≤ 0.7).

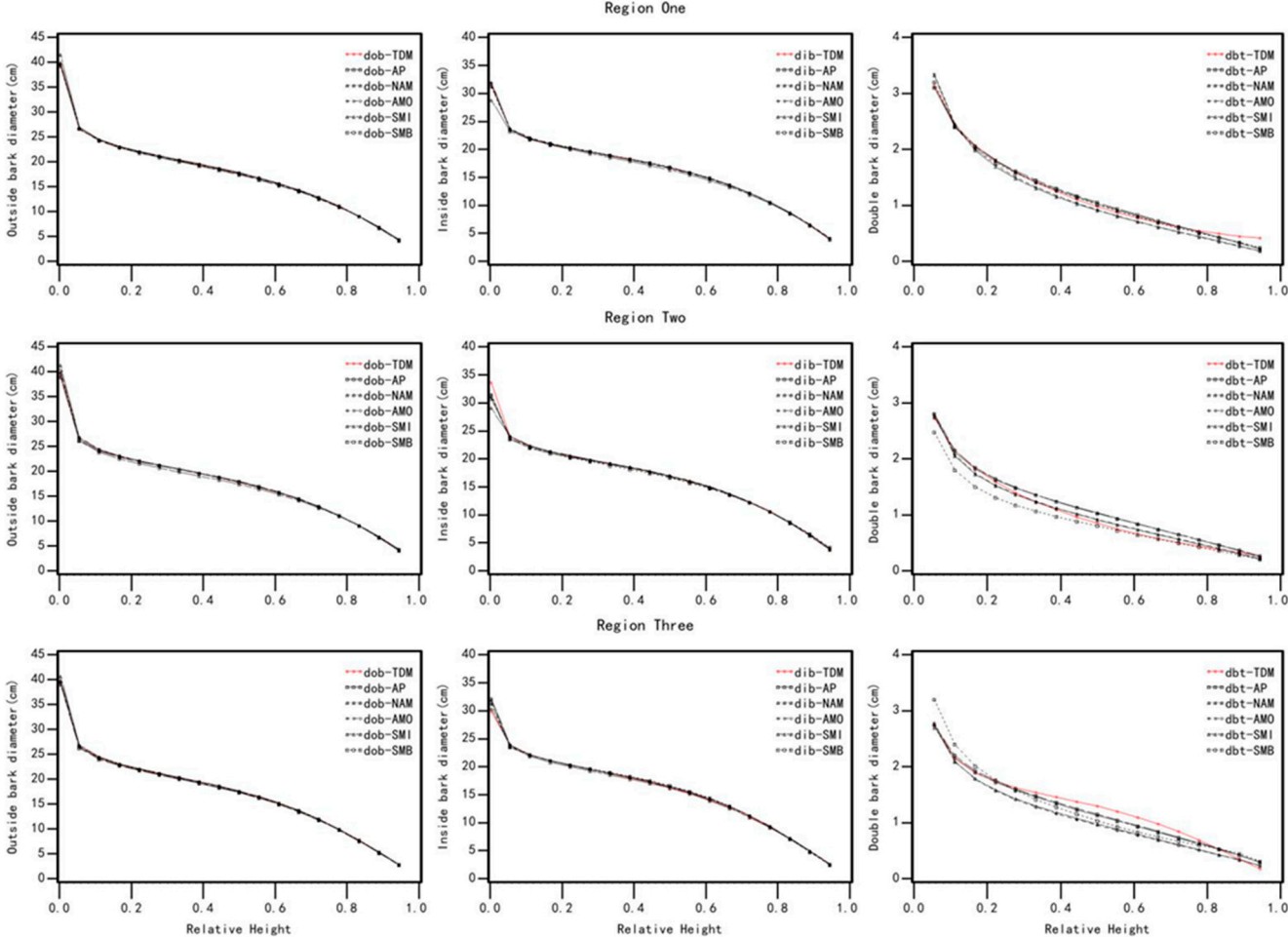

**Figure 6.** Predicted diameters over relative height using the studied methods for three regions (D = 26, H = 18).

## 4. Discussion

In this study, our data were collected across a large geographical region in Northeast China covering a variety of topographic and stand conditions. Six forms of nonlinear additive model structures were built and tested with the data. Our results demonstrated that the TDM model structure was nearly as accurate as other models in goodness-of-fit and predictive abilities in terms of average prediction errors for dob, dib and dbt. While other models had a slight advantage, differences between the six model structures were small. All model structures took into account the intrinsic correlations among the stem components and provided efficient parameter estimation. In NSUR model structures (i.e., AMO), the prediction accuracy of dob depended on the accuracy of dib and dbt models. The aggregation nature of the systems required that each component be estimated to obtain dob. A relatively large prediction error in any component model could affect the prediction accuracy of dob. The prediction precision of TDM and AP model structures depended on the accuracy of the dob model, which was the most accurate among the dob, dib and dbt models. The essence of TDM model structure is also the NSUR, which theoretically led to the unbiased estimation, satisfying the additive property of the total and components, and guaranteeing their proportions in the total model. It was also recognized that AMO, the NSUR method, had been widely used in additive volume or biomass equations throughout the world [32–34]. Admittedly, it was difficult to deduct and fit the additive system derived from the TDM. However, the advantages of TDM were substantiated in this study. These appealing characteristics of the TDM model structure were supported by our results. Therefore, the TDM was shown to be another better choice to develop additive

taper equations. This methodology would be useful for forest researchers interested in developing more precise systems of stem taper models in the future.

It is known that stem volume is important for forest management. Therefore, we further verified the prediction ability of six additive models for corresponding volume. The following Table 6 shows the model validation results of each part of volume calculated by numerical integration for each additive model. Comprehensive indicators and the precision of TDM were closely followed by AP for all variables. The two additive volume model methods are better than the other four models. They reflect the advantages of the additivity model and volume model constructed by TDM.

**Table 6.** Volume prediction for six additive models.

| Methods | Variable | $\bar{e}$ | $\delta$ | *RMSE* | *TRE* | $\sum rank$ |
|---|---|---|---|---|---|---|
| TDM | dob_v | 0.000335 | 0.000167 | 0.01293 | 0.39745 | 4.4 |
| | dib_v | −0.000194 | 0.000157 | 0.01252 | 0.42954 | 5.6 |
| | dbt_v | 0.000211 | 0.000009 | 0.00302 | 0.16405 | 7.1 |
| AP | dob_v | 0.000335 | 0.000167 | 0.01293 | 0.39745 | 4.4 |
| | dib_v | −0.000198 | 0.000158 | 0.01259 | 0.43433 | 9.1 |
| | dbt_v | 0.000214 | 0.000009 | 0.00301 | 0.16298 | 6.4 |
| NAM | dob_v | 0.001270 | 0.000169 | 0.01306 | 0.41129 | 13.5 |
| | dib_v | 0.000603 | 0.000158 | 0.01259 | 0.44350 | 14.4 |
| | dbt_v | 0.000349 | 0.000009 | 0.00305 | 0.16970 | 10.6 |
| AMO | dob_v | 0.000754 | 0.000166 | 0.01292 | 0.39974 | 6.6 |
| | dib_v | 0.000001 | 0.000158 | 0.01256 | 0.43505 | 6.8 |
| | dbt_v | 0.000435 | 0.000010 | 0.00316 | 0.18469 | 19.5 |
| SMI | dob_v | 0.000610 | 0.000166 | 0.01292 | 0.39866 | 5.7 |
| | dib_v | 0.000286 | 0.000159 | 0.01262 | 0.44226 | 12.6 |
| | dbt_v | 0.000005 | 0.000010 | 0.00319 | 0.17758 | 15.5 |
| SMB | dob_v | 0.000551 | 0.000180 | 0.01343 | 0.43078 | 20.2 |
| | dib_v | 0.000018 | 0.000164 | 0.01281 | 0.45253 | 19.1 |
| | dbt_v | 0.000215 | 0.000011 | 0.00325 | 0.19013 | 22.8 |

It should be noted that correlated error structure in the data was not taken into account in the model fitting process due to convergence problems. For instance, a test of auto-correlation for the TDM method showed that the models of dib and dbt failed to achieve convergence. Prediction accuracy is little affected by the correlated error structure [45]. For practical applications, auto-correlation is generally ignored when using models for prediction [46–50].

To date, additivity has been used in the forestry field, for calculations such as biomass, volume [12,13,32], and crown [10,11]. However, it is rare to study the additivity of taper equation. As stated earlier, only Rodríguez et al. [23] have conducted relevant research with the NSUR (SMB) method based on 351 Corsican pines, where the authors considered auto-correlation, and predicted the whole-tree volume and the different components of Corsican pine. Inspired by this, our paper constructed a new additive model, which compared the advantages and disadvantages of different existing additive structures. Our work expanded and perfected the additive theoretical system of taper equation.

## 5. Conclusions

This research is believed to be a novel attempt to present a preliminary additive system for Dahurian larch in Northeast China. Four approaches of TDM, AP, OLSSR, and NSUR were used to develop the additive systems of stem taper models for Dahurian larch. All model structures, particularly the TDM, demonstrated the additive property of stem taper models efficiently, with TDM obtaining a slightly better performance. In addition, the systems of stem taper models with the TDM method and dummy variable performed much better than those without the dummy variable. This methodology would be useful for forest researchers seeking to develop more precise systems of stem taper models, to predict volume in the future.

**Author Contributions:** Y.X. performed data analysis, and wrote most of the paper. M.K.S. assisted in data analysis and article revision. L.J. supervised and coordinated the research project, designed the experiment, and contributed to writing the paper. All authors have read and agreed to the published version of the manuscript.

**Funding:** This research was supported by Heilongjiang University of Science and Technology Research Start-up Fund granted to Yanli Xu.

**Acknowledgments:** We would like to acknowledge the efforts of all investigators, who took part in the field measured forest surveying of all data.

**Conflicts of Interest:** The authors declare no conflict of interest.

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
