# Peer review of "A Newly Built Model of an Additive Stem Taper System with Total Disaggregation Model Structure for Dahurian Larch in Northeast China"

_forests, doi:10.3390/f12101302_

Round 1

Reviewer 1 Report

Comments to the authors

I think in general the authors did a good job. The modeling approach is fairly thorough and appropriate. The research is practical. The study is relevant and adequate in light of present knowledge in taper modelling. The presentation of the manuscript is reasonably clear and focused. Providing the sas code in an appendix at the end of the manuscript for the presented total disaggregation model (TDM) would be very appreciated by potential readers.

I recommend the acceptance of this manuscript with minor revisions, as specified below and the comments done on the manuscript:

  • Page 6, lines 150-151: please improve the redaction of the following paragraph because it is not clear: ….The dib 150 and dbt were deduced and combined like terms.
  • Page 8, line 203: delete formula 15 and write directly the formula for R2a as it normally done in many papers.
  • Page 9, line 224: clarify which goodness-of-fit statistics have been used as Si.
  • Page 10, line 259: it should be table 4.
  • Page 16, line 289: it should be figures 3, 4 and 5.

Reviewer 2 Report

Manuscript (MS) has interesting topic but its significance could be greater if the MS have got another aim - comparison of results of the stem taper functions for the stem volume calculation. Knowledge about stem shape is important - it is true, but the stem volume is more important for forest management. That is the reason why I think that MS have to contain new parts in Data analysis and Results too. I recommend to authors that use their final models of stem taper equations and use it for volume calculation and compare errors of these volume models to each other. Only choice how to calculate volume from Kozak stem taper model is numeric integration. If authors will accept my recommendation the MS will be much better and its scientific soundness and interest to the readers will be better.

Detailed remarks:

Line 109: m is estimated parameter or fixed constant 0.1? It is confusing.

Equations 5, 6, 7: How was chosen optimal dependent combination of parameters r on bi? Please explain it better.

Lines 354-359: You wrote that in the models is problem with correlated error structure. You solved it by statement that convergence failed. It is not convenient. Especially if in the next paragraph of the Discussion wrote that Rodríguez  et al. (2013) solved this problem. Please try it again and remove the problem of auto-correlation.

Round 2

Reviewer 2 Report

All my recommendations were accepted and problematic parts of the MS were solved by authors. I do not have any other questions or remarks.